# MECHANISM-EMPOWERED MULTIVARIATE TIME SERIES FORECASTING MODEL: APPLICATION TO TUBERCULOSIS PREDICTION

## ABSTRACT

Among the current global health challenges, tuberculosis, as a highly contagious chronic disease, remains one of the major public health problems worldwide. Despite significant progress made in the past decades, new challenges, including systematic and effective downscaling, accurate prediction of disease incidence, and implementation of source reduction measures, have added to the difficulty of tuberculosis control. In view of the limitations of the recently proposed EIGHT prediction models in terms of prediction accuracy, this study adopts the Learnable Decomposition and Dual Focus Module Model (Leddam) and then introduces a novel mechanism-supported multivariate spatiotemporal series framework, termed LCHHA-Leddam, to address the challenges in tuberculosis forecasting through an investigation of coal power generation in China. This framework substantially simplifies the complexity of tuberculosis prediction, enhances accurate dimensionality reduction, and improves traceability. It also enhances the explanatory power and accuracy of the Leddam model in the field of tuberculosis prediction. This study provides a fresh perspective for enhancing epidemic forecasting and exploring source reduction measures for industrial activities, demonstrating the feasibility of AI-assisted public health strategies and green production.

## 1 INTRODUCTION

Climate change and environmental pollution have become one of the most important factors affecting public health with the acceleration of urbanization, industrialization, and economic development. Recent research has shown that environmental pollutants lead to approximately nine million premature deaths per year worldwide (Landrigan et al., 2018; Stanaway et al., 2018), and climate change is expected to cause approximately four million additional deaths per year (Zhao et al., 2021). Most studies have explored the relationship between diseases and environmental factor with respect to health risk (Asadgol et al., 2019, Danziger et al., 2022). Fewer studies comprehensively address the concurrent effects of climate change and environmental pollution on disease occurrence and implement source reduction measures. Risk factors for many human diseases and targets for preventive healthcare strategies are commonly identified based on epidemiological analysis (Chowdhury et al., 2018). However, a variety of factors/processes jointly affect the health outcome or its indicator, suggesting the inability of epidemiological models to simultaneously and systematically consider, and in many cases even impossible to solve hundreds of direct- and indirect environmental factors and processes (Braun et al., 2016).

Life cycle human health assessment (LCHHA) is a systematic and internationally standardized approach for simultaneously and efficiently quantifying the direct and indirect human health footprint of a targeted product, process, or activity from the beginning to the final phase (ISO 14040, 2006; ISO 14044, 2006). In contrast to epidemiological studies, LCHHA quantifies the potential human risk of a regional population via the dose-response curves derived from laboratory animal experiments and extrapolation factors for interspecies differences (Huijbregts et al., 2010). As animal experiments database-based LCHHA analysis has high uncertainty because of interspecies differences in physiology and metabolism, the combination of methods to understand better the sources' contribution to potential human health impacts is highly needed. Systematically and effectively implementing source reduction

measures and key factor identification to protect public health from environmental pollutants has recently become possible. For instance, industrial activity-environmental pollution-carcinogenic disease relationships calculated by LCHHA and epidemiological cancer incidences can now be further explored (Chen et al., 2019; Jia et al., 2022). However, relevant research on infectious disease and its prediction is extremely rare at the global level.

Interestingly, machine learning, as a popular approach for predictive modeling, has been extensively used in disease prediction to achieve early warning and disease intervention because of its capacities on analyzing large amounts of data, handling complex relationships, and learning and adapting over time (Huang et al., 2022; Krittanawong et al., 2020). Multivariate time series are datasets with multiple dimensions, where each dimension represents a separate univariate time series, e.g., climate characteristics (Zhang & Yan, 2023). The prediction of infectious diseases is a multivariate time series prediction as multiple factors/processes collectively influence infectious diseases. Predicting these time series helps to improve decision-making in various application areas such as weather, energy, and finance (Angryk et al., 2020; Demirel et al., 2012; Patton, 2013). Consequently, the incidence of infectious diseases can now be predicted, and the reduction of sources for human health protection can now be explored by integrating LCHHA, climate change, and multivariate time series forecasting models. Tuberculosis (TB) is used as a target disease to achieve the aforementioned goal; it is one of the world's leading infectious disease killers (WHO, 2022) and is involved in the 2030 Agenda for the World's Sustainable Development (United Nations, 2015). A quarter of the world's population is infected with Mycobacterium TB (Suárez et al., 2019). In 2021, the number of newly diagnosed TB patients in China was 0.78 million, ranking third among countries and accounting for 7.4% of global TB cases (WHO, 2022). In China, the incidence and death of TB rank first within the Class A and Class B notifiable infectious diseases. Two-thirds of the infected individuals are concentrated in the age group (15-70) with solid production capacity. Hence, the issue is regarded as a major public health problem in China.

TB transmission occurs almost exclusively because of aerosolized particles (Dinkele et al., 2022) and climate conditions such as temperature through the influence on TB bacteria spread and survival, human immune response, and climate patterns (Chong et al., 2022; Chormare and Kumar, 2022). However, the joint effects of climate change and environmental pollution on TB are also rarely assessed. As the world's largest coal consumer and producer, China is confronted with the serious environmental problem caused by coal burning (Ma et al., 2021), such as a global environmental issue of haze over China. More than half of greenhouse gas and aerosolized particles are produced from coal consumption in China (Ma et al., 2021; Shi et al., 2017). The coal power industry in China dominated national coal consumption since 2000 (>40% ), although renewable energy has been encouraged by the Chinese government (Chinese Energy Statistic Yearbook, 2001-2021). Approximately $5.3 \times 10^5$ premature deaths per year were recorded because of air pollution generated by coal burning in China (Yun et al., 2021). This research thereby focuses on the environmental pollutants released by the coal power industry in China and maps and predicts the incidence of TB caused by climate change and environmental pollution. Worth noting, TB incidence caused by climate factors and environmental pollutants is a complex and systematic phenomenon. Direct and indirect pollutants emitted from the coal power industry during its whole life cycle stages for LCHHA analysis generally involve hundreds of contaminants. Therefore, the intervention of low-dimensional feature engineering is highly needed. Creating low-dimensional features is crucial for data scientists to develop models and minimize model variance (Xuan et al., 2019; Rizgar et al., 2020). Domain experts typically undertake this feature engineering task manually, guided by specific domain knowledge. This method is known as mechanism-driven dimensionality reduction (MR) (Lan & Susan, 2019; Ting et al., 2010). Data-driven dimensionality reduction (DR), however, selects feature engineering methods based solely on available data. DR effectively extracts relevant features without relying on domain expertise, serving as a form of automated data cleaning (Neoklis et al., 2017). Despite this, MR remains essential since numerous parameters can significantly influence the performance of DR algorithms and modelling techniques, while MR can scientifically manage the relationships between parameters. This study compares MR and DR in terms of interpretability and accuracy, with LCHHA as a representative of MR. It also introduces a novel mechanism-enabled multivariate spatiotemporal sequence framework that effectively improves the accuracy and interpretability of infectious disease prediction and helps public health managers identify effective management and source reduction strategies.

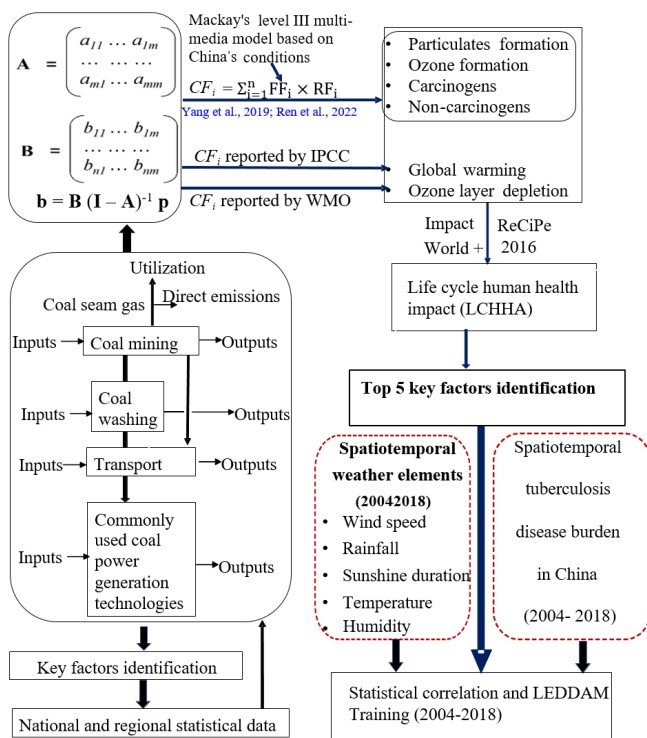

Figure 1: The prediction and source reduction approach of TB incidence caused by climate change and environmental pollution released by coal power generation.

## 2 METHODOLOGY

### 2.1 LCHHA OF COAL POWER GENERATION IN CHINA

The LCHHA of coal power generation in China was estimated based on ISO 14040 series standards (ISO 14040, 2006). Six subcategories (i.e., particulates formation, ozone formation, ozone layer depletion, global warming, carcinogens, and non-carcinogens) are involved for direct and indirect LCHHA (Fig.1). The selection of subcategories in this study is consistent with the commonly used LCHHA models established in Europe (Sala et al., 2012), North America (Bare, 2011), and World (Bulle et al., 2019). Regarding the global scale impact categories of ozone depletion and climate change, the characterization factors published by the World Meteorological Organization (WMO, 2011) and Intergovernmental Panel on Climate Change (IPCC, Arias et al., 2021) are used for quantification. For the rest subcategories, the SDU model, a life cycle impact assessment model developed for China's life cycle assessment analysis (Yang et al., 2019; Ren et al., 2022), is applied. The life cycle inventories of coal power generation at industry level are taken from previously published reference reported by our research group (Li et al., 2024 a, b). The system boundary is set to cradle-to-gate, and 1 kWh coal power generation is used as the functional unit. All inputs and outputs for each life cycle stage of coal power generation are utilized in this study except for infrastructure because of data limitations and their negligible impact (Ecoinvent Centre, 2016). The uses of LCHHA in this study are: 1) to evaluate the human health footprint of coal-fueled power industry initiatives at the national level; 2) to identify the key factors for MR and environmental improvement in China; 3) to provide a training set for predicting the TB incidence caused by direct environmental pollution released by the coal power industry in China; and 4) to enhance the precision and intelligibility of spatio-temporal sequence models. The formula of LCHHA is as follows,

$$\text{LCHHA} = \sum_{i=1}^{n} P_{q_i} \times \alpha_i \tag{1}$$

$$q_1 \begin{bmatrix} p_{11} \\ p_{21} \\ \vdots \\ p_{i1} \\ p_{i+1,1} \\ \vdots \\ p_{n1} \end{bmatrix} + q_2 \begin{bmatrix} p_{12} \\ p_{22} \\ \vdots \\ p_{i2} \\ p_{i+1,2} \\ \vdots \\ p_{n2} \end{bmatrix} + \ldots + q_m \begin{bmatrix} p_{1m} \\ p_{2m} \\ \vdots \\ p_{im} \\ p_{i+1,m} \\ \vdots \\ p_{nm} \end{bmatrix}$$

$$= \begin{bmatrix} p_{11} & p_{12} & p_{1m} \\ p_{21} & p_{22} & p_{2m} \\ \vdots & \vdots & \vdots \\ p_{i1} & p_{i2} & p_{im} \\ p_{i+1,1} & p_{i+1,2} & p_{i+1,m} \\ \vdots & \vdots & \vdots \\ p_{n1} & p_{n2} & p_{nm} \end{bmatrix} \begin{pmatrix} q_1 \\ q_2 \\ \vdots \\ q_m \end{pmatrix}$$

$$= Pq \tag{2}$$

$$\alpha_i = \mathrm{XF}_i \times \mathrm{EF}_i \times \mathrm{DF}_i = \left( \frac{\mathrm{BAF} \times \mathrm{PROD} \times \mathrm{POP}}{\mathrm{MASS}} + \frac{\mathrm{INH} \times \mathrm{POP}}{V_{\mathrm{air}}} \right)$$
$$\times \frac{0.5}{365 \times \mathrm{LT} \times \mathrm{BW} \times \mathrm{ED}_{50}} \times \mathrm{DF}_{\mathrm{NC/C},i}$$
$$+ \frac{\mathrm{EF}_{\mathrm{PF},i}}{\mathrm{EF}_{\mathrm{PM2.5}}} \times \mathrm{DF}_{\mathrm{PF}}$$
$$+ \frac{\mathrm{EF}_{\mathrm{OF},i}}{\mathrm{EF}_{\mathrm{NOx}}} \times \mathrm{DF}_{\mathrm{OF}}$$
$$+ \mathrm{EESC}_i \times \mathrm{UVB}_i \times \mathrm{EF}_{\mathrm{OD},i} \times \mathrm{DF}_{\mathrm{OD}}$$
$$+ \frac{\mathrm{GWP}_{\mathrm{GW},i}}{\mathrm{GWP}_{\mathrm{CO}_2}} \times \mathrm{DF}_{\mathrm{GW}} \tag{3}$$

where $P_{qi}$ and are the life cycle emission inventory ($\mathrm{kg}_{\mathrm{emission}}$) and health damage ($\mathrm{Daly/kg}_{\mathrm{emission}}$) of substance $i$. The meanings of rest symbols are listed in Table 3.

## 2.2 LCHHA-LEDDAM FRAMEWORK

The Learnable Decomposition and Dual Focus Module (Leddam) was selected for this investigation due to its notable enhancement in predictive precision compared to the recently identified EIGHT predictive models (Yu et al., 2024). Fig.1 presents the prediction and source reduction approach of TB incidence caused by climate change and environmental pollution released from coal power generation in China. The monthly spatiotemporal meteorological elements, the LCHHA of direct 66pollutants from coal power generation, and the epidemiological TB incidence in the 2004–2018 period are used for training the Leddam model. Spatiotemporal meteorological elements, including wind speed, rainfall, sunshine duration, temperature, and humidity data are selected as the indicators of climate change because these factors are measurable and easier to track changes over time.

$$\mathrm{Percentage}_i = \frac{P_{q_i} \times \alpha_i}{\mathrm{LCHHA}_{\mathrm{All}}} \times 100\% \tag{4}$$

$$X_{\mathrm{Seasonal}} = X_{\mathrm{Seasonal}}[i_1, i_2, i_3, i_4, i_5] \tag{5}$$

The analysis of the impact of LCHHA on Leddam begins with a comprehensive examination of the LCHHA effects associated with 34 categories of pollutants. The impact of each pollutant on LCHHA

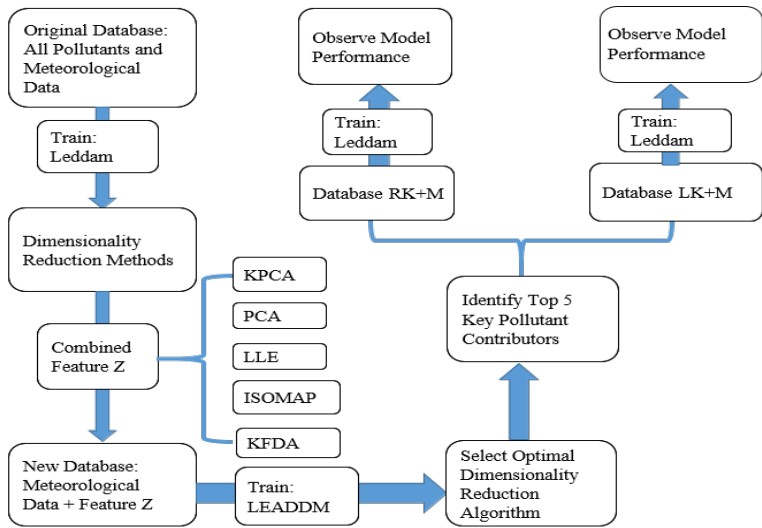

Figure 2: Flow chart of experiment

is quantified. The overall impact, designated as $LCHHA_{All}$, is determined by the summation of all pollutants' LCHHA impact. Index $i$ represents a distinct pollutant. Subsequently, the percentage contribution of each pollutant to the total is evaluated. These calculations are instrumental in developing an enhanced version of $X_{Seasonal}$, where $X_{Seasonal}$ of the top 5 contributed pollutants are indexed as $i_1$, $i_2$, $i_3$, $i_4$, and $i_5$.

## 2.3 DATA SOURCES

Climate data from 699 meteorological stations in China were obtained from the Meteorological Observation Information Centre of China to assess the relationship among various Meteorological elements (CMA, 2021). The spatiotemporal and annual average life cycle inventory of coal power generation was obtained from the CPLCID® (Zhang et al., 2016). For coal power generation in the CPLCID®, the data on spatiotemporal coal consumption for coal power generation, import and export coal amount, direct air emission amount of $SO_2$, NOx, and particulates, desulfurization gypsum and coal ash utilization rate, hospital beds, medical technicians, GDP per capita, population density, water consumption, and technological transformation were obtained from various reference documents (Annual Statistic Report on Environment in China, 2004-2021; China Electric Power Industry Annual Development Report, 2006-2020; China Statistical Yearbook, 2001-2021; Chinese Energy Statistic Yearbook, 2001-2021). The study by Hong et al. (2019) was also used to derive the data on national average methane and carbon dioxide emitted from low-, high-, outburst-, and open-pit coal mining sites and the life cycle inventory of coal-seam gas utilization. The data on coal quality at the national level were taken from Liu et al. (2015). The data on the primary inventory of coal-based electricity generation were taken from references (Cui et al., 2012; Xu et al., 2015; Zhao et al., 2015), and further combined with spatiotemporal regional statistical data according to the research of Hong et al. (2015). The data collection mentioned above were allowed for the building of a macro-level life cycle inventory of coal power generation in China (Li et al., 2024 a, b). Spatiotemporal TB incidence in China was obtained from China Health Statistical Yearbook (2004-2021) and Public Health Science Data Center (2023), whereas annual TB incidence and burden in China were taken from the Global Burden of Disease Study (IHME, 2020).

## 3 EXPERIMENT

Five downscaling algorithms were chosen based on the fact that dimensionality reduction can be categorized into two main types: linear mapping and nonlinear mapping. In this context, linear mapping was represented by Principal Component Analysis (PCA), while nonlinear mapping included kernel methods such as Kernel Principal Component Analysis (KPCA) and Kernel Fisher Discriminant

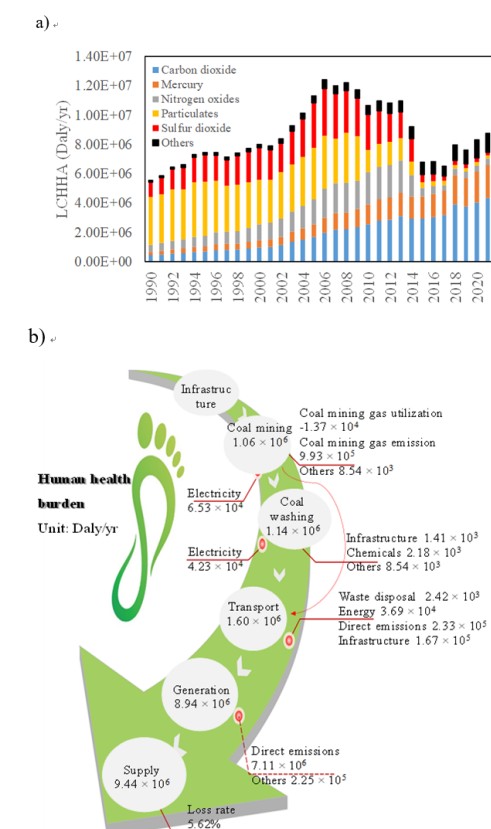

Figure 3: Contribution of dominant substances and processes to the annual LCHHA in China: a) substance; b) process

Analysis (KFDA), as well as manifold learning techniques like Isometric Mapping (ISOMAP) and Locally Linear Embedding (LLE). The training sets include spatiotemporal annual epidemiological TB incidence, LCHHA values, and meteorological elements at China's provincial level in the 2004–2018 period (n=5580). Epidemiological TB data involve spatiotemporal reported incidence rate, incidence cases, and disease burden. Spatiotemporal meteorological elements including wind speed, rainfall, sunshine duration, temperature, and humidity data are selected as the indicators of climate change because these factors are measurable and easier to track changes over time. Spatial average monthly meteorological elements at the province level integrated by daily surface climate data of 699 meteorological stations in China since 2004 are used as original database. Data are randomly split and set into a training, validating and testing set with a ratio of 7:1:2. The mean squared error loss (MSE), mean absolute error (MAE), and coefficient of determination (i.e., R-squared) are used for aforementioned models screening. Fig.2 illustrates the process of our experiment. The LEADDM was initially trained with a database containing all pollutants and meteorological variables. KPCA, PCA, KFDA, LLE, and Isomap were applied to the pollutant data, creating a feature set Z. A new database, combining meteorological data and Z, was then used to retrain the LEADDM. The best algorithm, determined by $R^2$, MSE, and MAE, identified the top five pollutant contributors, results shown in Table 1. These, along with meteorological data, formed the RK+M[DM1] database. Similarly, LK+M was created using LCHHA analysis. Both databases were used for further Leddam training.

## 4 RESULTS

### 4.1 ANALYSIS OF KEY CONTRIBUTORS THROUGH LCHHA

Fig.3 presents the most significant substances and processes contributing to the LCHHA. Mercury, carbon dioxide, nitrogen oxides, sulfur dioxide, and particulates generated by coal power generation

Table 1: Selection of dimensionality reduction method based on the database (P=All pollutants, M=Meteorological)

|  | P+M+PCA | **M+KPCA** | M+LLE | M+ISOMAP | M+KFDA |
|---|---|---|---|---|---|
| Test-$R^2$ | 0.753 | **0.767** | 0.560 | 0.612 | 0.650 |
| Test-MSE | 0.142 | **0.264** | 0.273 | 0.119 | 0.261 |
| Test-MAE | 0.266 | **0.143** | 0.266 | 0.228 | 0.337 |

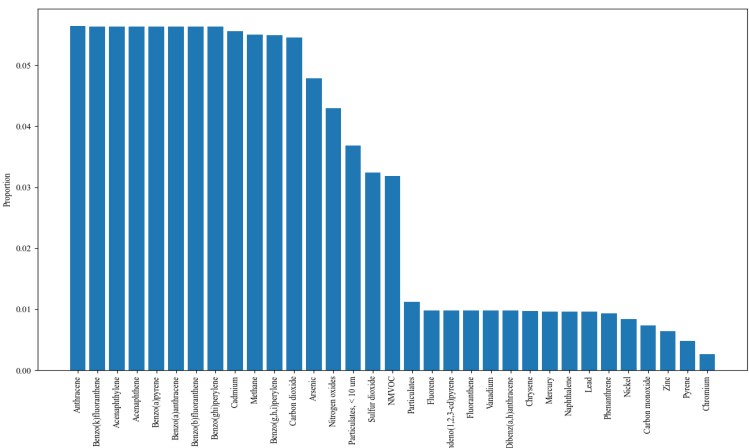

Figure 4: Each pollutant proportion in KPCA component

site appear to be the main contributors to the overall impact on human health. The potential impact generated by the rest of the substances (e.g., PAHs, lead, arsenic, zinc, and VOCs) and processes (e.g., wastewater and solid waste disposal, limestone, oil, hydrochloric acid, sulphuric acid, sodium hydroxide, coal washing, transportation, and raw materials consumed during coal mining stage) is low. Consequently, the pollutants emitted directly from the coal power generation stage are utilized for subsequent study. The overall LCHHA of coal power generation across China over 30 years has significantly decreased in the LCHHA of nitrogen oxides, sulfur dioxide, and particulates for the last decade, whereas an increasing tendency is observed for the other key substances (Fig.3a). Strict air pollution demands by the Chinese government (GB 13223-2011) and the annual increase in coal power generation can explain the variations.

### 4.2 ANALYSIS OF KEY CONTRIBUTORS THROUGH ML DIMENSIONALITY REDUCTION METHODS

Table 1 presents the performance of various dimensional reduction techniques applied to the pollutant meteorology dataset. The results indicate that Kernel PCA (P+M+KPCA) achieved the highest Test R² of 0.767, a lowest Test MAE of 0.266 and not the highest Test MSE of 0.264, demonstrating strong capability in capturing different pollutants relationships. KPCA is selected to do the analysis of key contributors. Fig.4 demonstrates the proportion of different pollutants in the KPCA component. Acenaphthene, Acenaphthylene, Anthracene, Benzo(a)pyrene, and Benzo(k)fluoranthene are the top five contributors to KPCA component, which is different from the analysis through LCHHA. Table 4 and Fig.8-11 in the appendix show the proportion of the various contaminants present in the alternative downscaling techniques. It is notable that no two of the five downscaling techniques yield an identical ranking of the first five key contributors, which suggests that the data-driven method lack sufficient interpretability. This phenomenon also demonstrates the limitations of existing dimensionality reduction methods in the field of AI for science.

### 4.3 COMPARISON BETWEEN MR AND DR

The results in Table 2 highlight the comparative performance of the different mechanisms in Leddam. The model that incorporates all pollutants along with meteorological data (P+M) exhibits the lowest

Table 2: Comparison between MR and DR (P=All pollutants, M=Meteorological, LK=Key factor selected by LCHHA, RK=Key factor selected by KPCA)

|            | P+M     | KPCA+M | **LK+M** | RK+M  |
|------------|---------|--------|----------|-------|
| Test-$R^2$ | -50.624 | 0.767  | **0.882** | 0.794 |
| Test-MSE   | 0.083   | 0.264  | **0.096** | 0.146 |
| Test-MAE   | 0.150   | 0.143  | **0.185** | 0.253 |

test $R^2$ of -50.624, suggesting the need to select key pollutants. In contrast, the model based on meteorology and LCHHA selection on pollutants (LK+M), has the highest test $R^2$ of 0.882. It also has the lowest Test MSE of 0.096 and lower Test MAE of 0.185 compared with KPCA selection (RK+M) and KPCA reduction (KPCA+M). Notably, TB incidence caused by climate factors and environmental pollutants is a complex and systematic phenomenon. LCHHA quantifies all the inputs and outputs of coal power generation during its life cycle stages. Every single input has its life cycle and interacts as a part of the coal power generation system. Therefore, LCHHA analysis can significantly reduce the complexity of the integrated assessment model applied in this study. The performance of LK+M underscores the effectiveness of the mechanism in enhancing Leddam model accuracy. Compared to data-driven methods, the higher accuracy of key factor locking via mechanism models highlights deficiencies of data-driven methods in exploring feature interrelationships. It also reveals that the data-driven downscaling algorithms still face multiple limitations, such as challenges in handling high-dimensional sparse data, the the effectiveness of feature extraction, and the inherent complexity of scientific problems. These limitations not only affect the depth and breadth of data analysis but may also restrict the process of scientific discovery.

## 4.4 SOURCE REDUCTION

Fig. 3 shows that carbon, mercury, and particulate matter (e.g., nitrogen dioxide, sulfur dioxide. particulates) emitted from coal burning for power generation are the main contributors of the direct LCHHA, which is the key contributor to TB prediction (Fig.1 and Table 2). The uptrend of life cycle carbon emissions from China's coal power industry and actual annual average temperature (Appendix A.1) are consistent with the IPCC opinion that global warming is mainly caused by greenhouse gas emissions (Arias et al., 2021). As the annual average temperature increase in China correlates with the annual humidity, rainfall, and sunshine duration changes (Fig. 5b), source reduction measures on TB become possible through pollutants control, specifically on mercury, carbon dioxide, nitrogen oxides, sulfur dioxide, and particulate emissions control. China continues to be the largest carbon (IEA, 2022) and mercury (Hu et al., 2018) emitter in the world because of its high gross national coal consumption. However, significant environmental control achievements have been attained in recent years (Fig. 3 a). Approximately 2.2 Mt of NOx, 0.8 Mt of $SO_2$ and 643.1 t of atmospheric Hg were emitted by the entire coal power generation chain in 2021. For the same year, in the entire coal power generation chain, approximately 40.1% of NOx, 69.9% of $SO_2$, and 98.0% of Hg were emitted from the coal power generation stage; the direct emissions of the aforementioned key substances by coal power generation sites in China in 2013 and the extensive haze episodes in China were nearly 89.8%, 96.6%, and 97.8%. The results indicate significant environmental control achievements in national $SO_2$, PM, and NOx control after setting a strict emission standard for thermal power plants (GB 13223-2011) and increasing the efficiency of coal consumption year by year (Chinese Energy Statistic Yearbook, 2001-2021). A similar trend of carbon control through coal consumption efficiency improvement can also be seen in Appendix Fig.12. However, mercury control is weak, and the current mercury mitigation policy in China's coal power industry remains unsystematic and ineffective. The USEPA (1977) reported that 60% of mercury could be removed using coal-washing technology, while the United Nations Environment Programme reported it to be 78% (UNEPA, 2002). Thus, the proper selection of coal-washing technology can significantly reduce the key factors and overall environmental burden. In 2020, the coal-washing rate in China was approximately 74.1%, lower than that in the United States (90%) (Ghosh, 2013). Clearly, increasing the national coal-washing rate and the efficiency of coal consumption is another key factor for protecting human health.

## 5 CONCLUSION

This study employs a novel LCHHA-Leddam model framework to predict the incidence of TB in China, taking into account spatiotemporal climate change and environmental pollution caused by the country's coal power industry. The LCHHA model can identify the connections between industrial activities, direct and indirect environmental pollution, associated risks, and their impact on human health, thereby facilitating the identification of source control measures. The results demonstrate that the LCHHA-Leddam model outperforms the traditional Leddam model due to its superior accuracy in identifying key factors through mechanism models. This suggests that a combination of mechanistic and data-driven models can significantly enhance the scientific accuracy of TB disease predictions. Given that the LCHHA of coal power generation in China is primarily characterized by direct emissions of atmospheric mercury, nitrogen oxides, sulfur dioxide, and particulate matter during the coal combustion stage for power generation, it becomes feasible to implement source reduction measures for TB disease control. These measures include controlling the aforementioned pollutants, improving coal consumption efficiency, and increasing the national coal washing rate.

In summary, this study's novelty lies not only in providing scientific data for early TB warning but also in its potential to facilitate effective management and source reduction. To improve the accuracy of predicted outcomes and the efficacy of public health protection measures, it is necessary to consider other factors that may influence epidemic diseases, such as high spatial-temporal resolution, genetics, diet, lifestyle regularity, and population distribution characteristics. Furthermore, incorporating interpretable machine learning algorithms, such as SHAP (SHapley Additive exPlanations), into our framework could enhance interpretability and provide a more comprehensive understanding of the factors influencing epidemic predictions. This holistic approach will better equip us to address the environmental and health challenges of the future.

## 6 REPRODUCIBILITY STATEMENT

To foster reproducibility, we will make our code available in supplementary materials and will make it public online after acceptance. We give details on our experimental protocol in Appendix

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

# A  APPENDIX

Table 3: LCHHA formula explanation

| Parameter | Meaning |
|---|---|
| $XF_i$ | Exposure factor of substance i |
| $EF_i$ | Effect factor of substance i |
| NC | Human health damage (DLAY) caused by carcinogenicity |
| C | Human health damage (DLAY) caused by non-carcinogenicity |
| PF | Human health damage (DLAY) caused by particulate formation |
| OF | Human health damage (DLAY) caused by ozone formation |
| OD | Human health damage (DLAY) caused by ozone depletion |
| GW | Human health damage (DLAY) caused by global warming |
| BAF | Bioconcentration factor |
| PROD | Intake per unit time |
| MASS | Total mass of pollutants in the exposure area |
| INH | Average respiratory rate per person |
| Vair | Total air volume in the region |
| LT | Average life expectancy of the population |
| BW | Average body weight of the population |
| $ED_{50}$ | Median effective dose: the dose that elicits the desired effect in 50% of subjects |
| $EF_{PF,i}$ | Health impact per kilogram of inhaled substance i |
| $EF_{PM2.5}$ | Health impact per kilogram of inhaled PM2.5 |
| $DF_{PF}$ | Human health damage caused by per unit intake of PM2.5 |
| $EF_{OF,i}$ | Health impact per kilogram of inhaled substance i |
| $EF_{NOx}$ | Health impact per kilogram of inhaled NOx |
| $DF_{NOx}$ | Human health damage caused by per unit intake of NOx |
| UVB | Increase in radiation in the area |
| $EF_{OD}$ | Additional disease incidence rate due to increased UV radiation in the area |
| $DF_{OD}$ | Human health damage caused by disease incidence due to UV radiation |
| $GWP_{GW,i}$ | Global warming potential of greenhouse gas i as reported by the latest IPCC |
| $GWP_{CO_2}$ | Global warming potential of CO2 as reported by the latest IPCC |
| AGWP | Absolute global warming potential in yr·W·m$^{-2}$·kg$^{-1}$ |
| TF | Temperature factor in C·m$^2$·W$^{-1}$ |
| RR | Relative health risks from rising temperatures |
| $DF_{GW,i}$ | Damage factor for health damage caused by emissions of greenhouse gas i, calculated through AGWP, TF, RR, and DALY·yr$^{-1}$·°C$^{-1}$ |

## A.1  CLIMATE CHANGE IN CHINA

Carbon dioxide, the main driver of climate change, has Carbon dioxide, the main driver of climate change, has attracted global attention. In China, carbon dioxide emitted from the whole country (Friedlingstein et al., 2022) and its life cycle emissions from the coal power industry show a significant uptrend during the last seven decades. The life cycle carbon emissions of the coal power industry currently account for half of the national carbon emission in China. Meanwhile, the annual average temperature in China shows a sharp rise (Fig.5a), with a warming rate of 0.25 °C/10 years, which is higher than the global average level for the same period (0.15°C to 0.20 °C/10 years; NASA, 2022). Spatiotemporal average humidity and rainfall show an uptrend with increasing average temperature and a downtrend with increasing sunshine duration, whereas annual average wind speed with temperature below -2.5 °C presents a downtrend (Fig.5b). For spatiotemporal wind speed beyond -2.5 °C, no significant changes are observed with increasing annual average temperature. The average rainfall distribution in China shows a decreasing tendency from the southeast coast to the northwest inland (Fig.5c). A similar tendency can be observed in the average humidity distribution, as the humidity distribution is closely related to the precipitation distribution. By contrast, the annual average sunshine duration shows an increasing tendency from the southeast coast to the northwest inland. The average temperature distribution is high in the south and low in the north, whereas a contrasting phenomenon is observed in the average wind speed distribution.

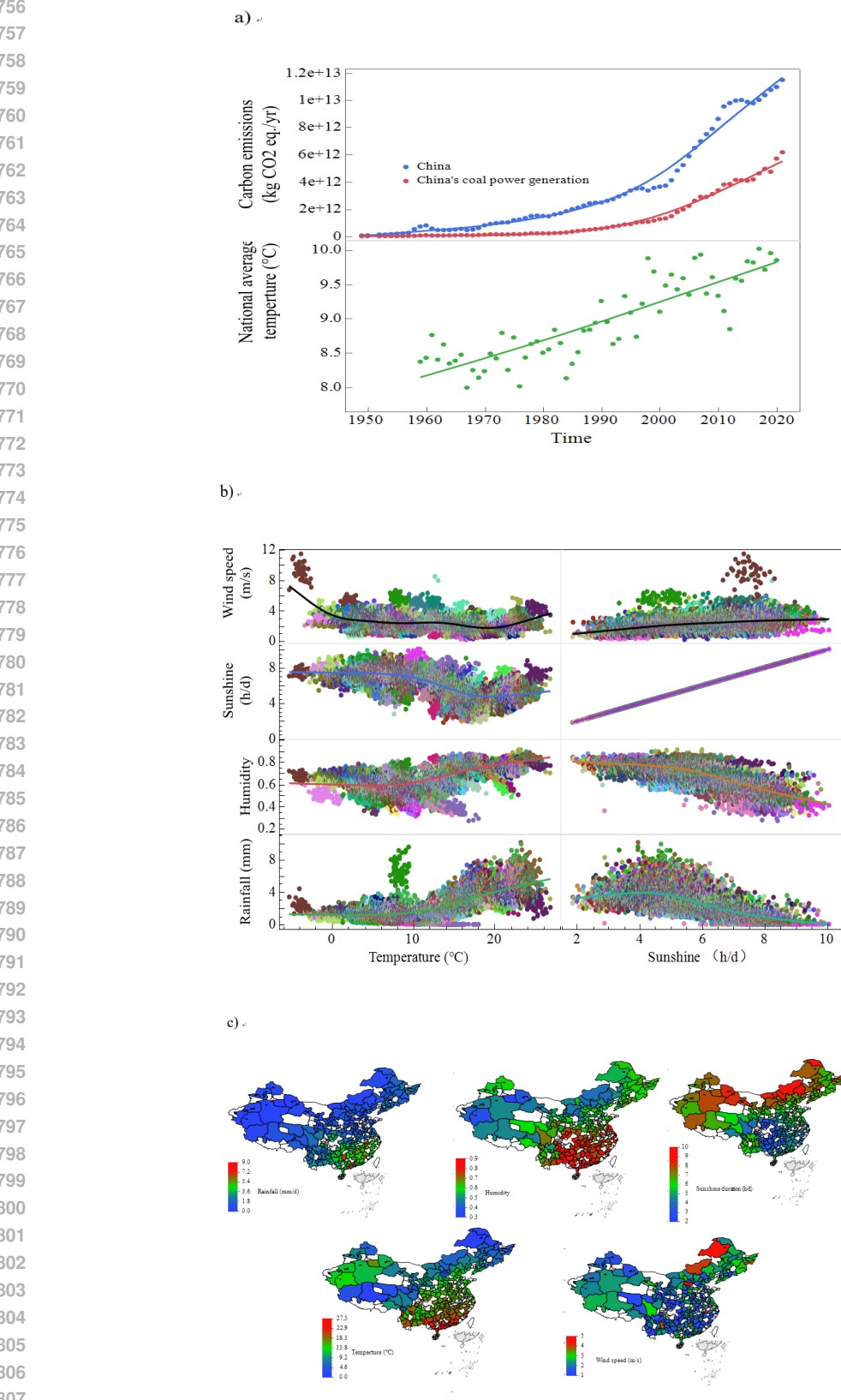

Figure 5: Spatiotemporal meteorological element changes: a) annual average variation since 1951; b) relationship among meteorological elements since 1951 (color represents different region); c) spatial variation in 2019 (white color means no data).

a)

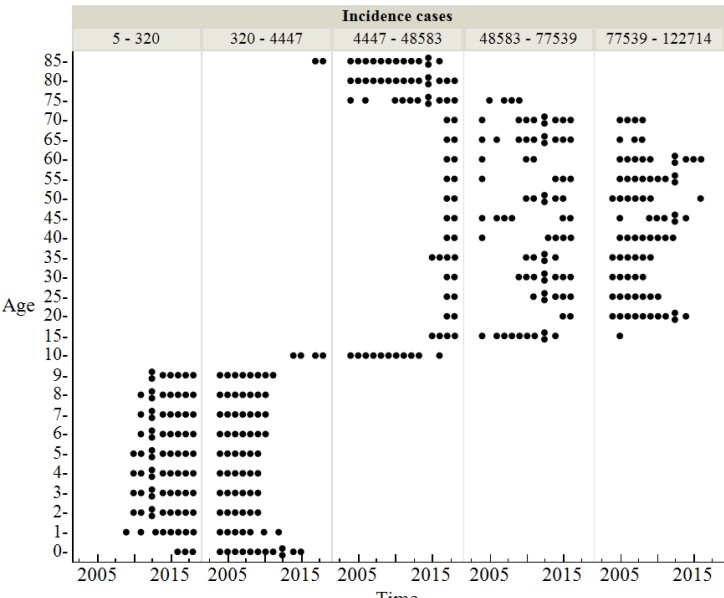

b)

Figure 6: Relationship between age and tuberculosis incidence in China a) incidence rate; b) incidence cases

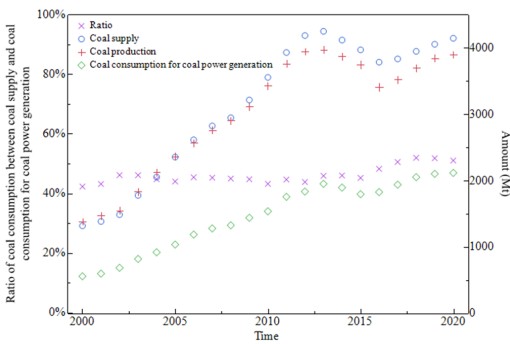

Figure 7: Secular variation of China's coal supply, coal production, and coal consumption in coal power industry.

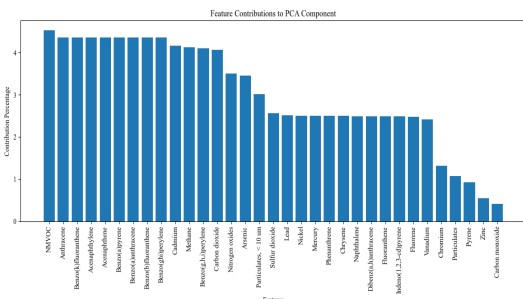

Figure 8: Each pollutant proportion in PCA component

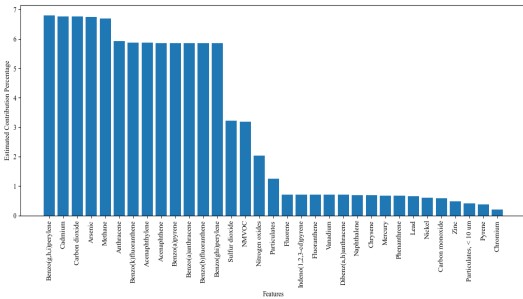

Figure 9: Each pollutant proportion in LLE component

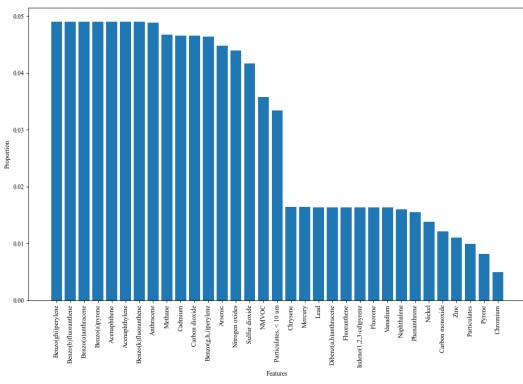

Figure 10: Each pollutant proportion in KFDA component

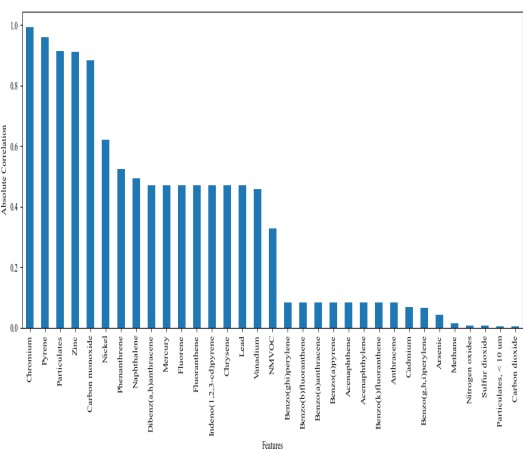

Figure 11: Each pollutant proportion in ISOMAP component

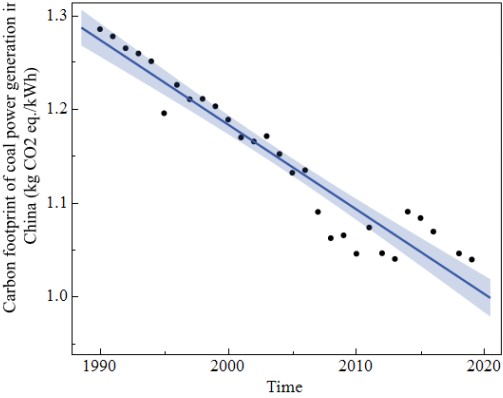

Figure 12: Carbon footprint of coal power generation in China during last three decades.

Table 4: Top 5 Key Contributors for 5 Different DR Methods

| PCA Feature | LLE Feature | KPCA Feature | KFDA Feature | ISOMAP Feature |
|---|---|---|---|---|
| NMVOC | Benzo(g,h,i)perylene | Anthracene | Benzo(ghi)perylene | Chromium |
| Anthracene | Cadmium | Benzo(k)fluoranthene | Benzo(b)fluoranthene | Pyrene |
| Benzo(k)fluoranthene | Carbon dioxide | Acenaphthylene | Benzo(a)anthracene | Particulates |
| Acenaphthylene | Arsenic | Acenaphthene | Benzo(a)pyrene | Zinc |
| Acenaphthene | Methane | Benzo(a)pyrene | Acenaphthene | Carbon monoxide |

Table 5: Parameter values

| Parameter | Value |
|---|---|
| seq_len | 96 |
| pred_len | 1 |
| d_model | 256 |
| n_layers | 3 |
| learning_rate | 1.00E-04 |
| kernel_size | 25 |
| features | MS |
| target | OT |
| freq | m |
| label_len | 48 |
| enc_in | 11 |
| dec_in | 11 |
| c_out | 11 |
| pe_type | no |
| dropout | 0 |
| revin | TRUE |
| num_workers | 0 |
| itr | 1 |
| train_epochs | 100 |
| batch_size | 32 |
| patience | 6 |
| lradj | constant |
| use_amp | TRUE |
| use_gpu | TRUE |
| gpu | 0 |

