# OpenReview forum: "Mechanism-Empowered Multivariate Time Series Forecasting Model: Application to Tuberculosis Prediction"
_ICLR.cc/2025/Conference — ICLR 2025 Conference Withdrawn Submission_

### Official Review · Reviewer_d6DJ · 2024-11-03

**Soundness:** 2
**Presentation:** 2
**Contribution:** 2
**Rating:** 3
**Confidence:** 4

**Summary:**

The paper introduces a novel "LCHHA-Leddam" framework to predict tuberculosis (TB) incidence in China by combining life cycle human health assessment (LCHHA) with the Learnable Decomposition and Dual Focus Module (Leddam) model. This framework integrates environmental and climate variables, specifically examining pollutants from China's coal industry, to create a multivariate time series model for TB prediction. By focusing on dimensionality reduction and mechanism-based approaches, the model improves interpretability and accuracy in TB forecasting, offering insights into effective pollutant control measures to mitigate public health impacts.

**Strengths:**

Some of the key strengths of the paper are:
1. The authors proposed an innovative Integration of Mechanistic and Data-Driven Models for infectious disease predictions. Specifically, the combination of LCHHA with Leddam enhances interpretability and accuracy, showcasing a promising hybrid approach for infectious disease prediction.
2. The authors also addressed a key issue - specifically, by focusing on TB and linking it to environmental pollutants from coal power, the study addresses a critical health issue and offers practical implications for policy-making in pollution control.
3. Finally, the authors provided a holistic view of the problem setup by considering diverse data sources. The framework incorporates extensive climate, epidemiological, and pollution data, which strengthens its predictive validity and offers a detailed methodology for future research.

**Weaknesses:**

However, given all of these there are some key shortcomings in the paper.
1. The authors provided an interesting case study for TB. However, when considering the broad field of infectious disease modeling/prediction, it is not evident how transferrable the framework is to other diseases or regions.
2. The motivation behind combining these methodologies and its implications for broadening epidemic forecasting remain vague, potentially limiting its contribution beyond this case study. For example, beyond the proposed combination, it is not evident what is the key shortcomings of the current landscape of the methods that this approach splves
2. Finally, the complexity of the implementation is not discussed sufficiently. For example,  the use of LCHHA and Leddam together with detailed pollutant tracking adds complexity, which may challenge reproducibility and scalability for broader public health applications.

Beyond the above there are a few minor issues in the paper
- There are many terms that have been presented without context. For example, Pg 3, Line 151, what is cradle-to-gate?
- Similarly there are many formulae presented without completely describing the terms. For example, Formula 3, what are BAF and so on
- Also the presentation in the paper can be improved upon. In the abstract it seems that the authors claim that improved disease prediction has created a challenge - this seems counter-intuitive

**Questions:**

See weakness above

---

### Official Review · Reviewer_ptec · 2024-11-03

**Soundness:** 2
**Presentation:** 2
**Contribution:** 1
**Rating:** 3
**Confidence:** 4

**Summary:**

In this work, the authors utilise a time series prediction model to predict the tuberculosis prediction.

**Strengths:**

The conclusion might has some clinical relevancy.

**Weaknesses:**

This paper clearly does not align with the scope of ICLR and has limited relevance to the spatio-temporal and time series prediction community. Its contributions may be more suited to a domain-specific journal. The work introduces no novel methodology; adding the LCHHA appears to be an incremental feature addition rather than a model innovation. Additionally, it is unclear how spatial patterns are incorporated into the model. Furthermore, the baseline models lack basic modern time series approaches (e.g., LSTM, RNN, ARIMA) and do not include any established spatio-temporal prediction models.

**Questions:**

Please refer to weaknesses.

---

### Official Review · Reviewer_LNjT · 2024-11-04

**Soundness:** 2
**Presentation:** 3
**Contribution:** 2
**Rating:** 3
**Confidence:** 4

**Summary:**

The paper proposed a mechanism-supported multivariate time series forecasting model on Tuberculosis prediction. The proposed method was evaluated on a single real-world dataset to forecast tuberculosis cases from coal power generation and climate features, with Learnable Decomposition and Dual Focus Module Model (Leddam).

**Strengths:**

- Originality: the paper is novel in terms of comparing mechanism-generated features with data-driven features in the forecasting utility.
- Quality: multiple evaluation metrics are reported and compared across different feature combinations, and feature reduction approaches.
- Clarity: the paper did a good job in addressing the importance of climate changes towards disease prediction and illustrating feature generation context.

**Weaknesses:**

Technical soundness is somehow weak in this paper:
- Though multiple metrics were reported, it's unknown that whether the results are cross-validated or not given no confidence intervals were provided. Predicted vs true values, and residual plots are necessary for model selection and comparison.
- Since it's a time series forecasting, it's unknown how the model performs across time. Also plots might be helpful to examine the forecasting utility.
- The paper focused more on feature generation and comparison and lacks baseline forecasting algorithms. A different regressor or forecasting model might generate better performance with different feature sets.

**Questions:**

- How would the proposed method generate to other countries' dataset?

---

### Note · Authors · 2024-11-14

**Comment:**

The reviews do not seem to welcome applied research.

**Withdrawal Confirmation:**

I have read and agree with the venue's withdrawal policy on behalf of myself and my co-authors.